# Role of Membrane Technology in Absorption Heat Pumps: A Comprehensive Review

**DOI:** 10.3390/membranes10090216

**Published:** 2020-08-31

**Authors:** Jonathan Ibarra-Bahena, Shankar Raman, Yuridiana Rocio Galindo-Luna, Antonio Rodríguez-Martínez, Wilfrido Rivera

**Affiliations:** 1Instituto Mexicano de Tecnología del Agua, Paseo Cuauhnáhuac 8532, Colonia Progreso, Jiutepec 62550, Morelos, Mexico; jibarra@ier.unam.mx; 2Instituto de Energías Renovables, Universidad Nacional Autónoma de México, Privada Xochicalco S/N, Col. Centro, Temixco 62580, Morelos, Mexico; shankar@ier.unam.mx (S.R.); ygalu@ier.unam.mx (Y.R.G.-L.); 3Centro de Investigación en Ingeniería y Ciencias Aplicadas, Universidad Autónoma del Estado de Morelos, Av. Universidad 1001, Col. Chamilpa, Cuernavaca 62209, Morelos, Mexico; antonio_rodriguez@uaem.mx

**Keywords:** membrane, absorption heat pump, absorber, desorber, heat and mass exchanger, cooling, heating

## Abstract

The role of heat pumps is linked to the actions of human life. Even though the existing technologies perform well in general, they have still some problems, such as cost, installation area, components size, number of components, noise, etc. To address these issues, membrane technologies have been introduced in both heat and cooling devices. The present work proposes and studied the review of the role of membrane technology in the heat pumps. The study focuses on the advancement and replacement of membrane in the place of absorption and compression heat pump components. The detailed analysis and improvements are focused on the absorber, desorber, and heat and mass exchanger. The parameters conditions and operation of membrane technologies are given in detail. In addition to this, the innovation in the heat pumps using the membrane technology is given in detail.

## 1. Introduction

A membrane is a porous barrier that separates two homogeneous phases of fluids by restraining totally or partially the transport of one or more chemical species present in a homogeneous fluid mixture. Transport through the membrane occurs when a driving force exists between these phases. The processes involving membranes are classified according to the driving force used, such as pressure difference (e.g., reverse osmosis, nano and ultrafiltration), concentration difference (e.g., dialysis and gas separation), electrical potential difference (electrodialysis), and partial pressure difference (membrane distillation) [1]. Membrane technology shows several advantages: Simplicity in operation, high selectivity for specific chemical species, easy to modularize and scale-up, reduction of chemical pre-treatments in the feed solutions and low waste generation, and low energy consumption [2]. For these reasons, the use of membrane modules and membrane contactors has been broadly used in many industrial sectors, especially in the petrochemical industry for hydrogen recovery [3,4], for treatment of textile industry waste-water [5,6], in the production of many medicines and products in pharmaceutical and biotechnology industry [7,8], in food production sectors [9,10], in seawater and brackish water desalination [11,12], and many others [13]. One of the new applications of the membranes devices is in absorption heat pumps.

An absorption heat pump refers to thermodynamic cycles able to transfer heat from a low to a high temperature. This category includes absorption chillers and absorption heat transformer systems, and both operate with a working mixture integrated by a refrigerant fluid and an absorbent [14]. The simplest absorption chiller configuration includes a desorber, an absorber, a condenser, an evaporator, an expansion valve, a throttling valve and a pump; additionally, a solution heat exchanger is included in order to increase the coefficient of performance (COP) [15]. The cycle is as follows: A constant heat flux is supplied to the generator until the boiling point of the mixture is reached in order to separate a fraction of the refrigerant fluid in the vapor phase at a high relative pressure and temperature. This vapor is condensed inside the condenser at a temperature close to the environment. Then, the refrigerant in the liquid phase passes through the expansion valve and is evaporated inside the evaporator at low pressure and temperature, producing the cooling effect. The refrigerant fluid in the vapor phase passes to the absorber where it is absorbed by the concentrated in absorbent solution pumped from the desorber. Because the absorption process of the refrigerant vapor in the absorbent-solution is exothermic, heat is produced—which has to be dissipated to the environment. The diluted in absorbent solution, resulting from the absorption process is lead to the desorber, passing through the throttling valve starting the cycle again. A solution heat exchanger (SHE) is placed between the desorber and the absorber in order to recover the heat from the hot solution stream reducing the desorber heat load required and enhancing the system COP. Figure 1a shows the P-T diagram of an absorption cooling cycle.

An absorption heat transformer (AHT) cycle is similar to the cycle described previously, but the goal is to produce a useful heat in the absorber at higher temperature level than the heat supplied to the system. A constant heat load is supplied to the desorber and the evaporator, at a relatively moderate temperature. The refrigerant fluid is separated inside the desorber, and it is condensed in the condenser, but it is evaporated inside the evaporator at a relatively high pressure. The absorption process is carried out at the highest pressure and temperature, and the useful heat produced by the absorber can be used in other processes to reduce the thermal energy requirements. Figure 1b shows the P-T diagram of an absorption heat transformer cycle.

The current research of the absorption heat pumps is toward to miniaturization of the components for small scale duty, such as single-family houses, small buildings, or automotive applications [16,17]; however, the conventional heat exchangers used as desorbers, heat and mass exchangers, and desorbers, restrict adopting the absorption heat pump in these fields because are big in size and heavy. In this respect, the membrane devices are a promising technology to surmount the disadvantages of the conventional components. The use of membrane contactors in absorption heat pump systems has been proposed and studied because these devices provide “artificial” interphase between two fluids [18]; therefore, membrane contactors are suitable for the desorption, absorption and heat and mass recovery processes. The aim of the present paper is to show the role, importance, advancement, and replacement of membrane technology in absorption heat pumps. Hence, a detailed review is carried out on heat pumps in terms of membrane-based components, such as desorbers, absorbers, and heat and mass exchangers.

## 2. Membrane-Based Desorbers

In absorption heat pumps, desorption is the separation process of the refrigerant fluid from the working mixture (integrated by a refrigerant fluid and an absorbent), and the component where the process takes place is called “desorber” (also known as generator). The most used desorber configurations are: Nucleate pool-boiling [19,20,21,22,23,24] and falling-film [25,26,27,28,29]. In the nucleate pool-boiling configuration, a constant heat load must be added in order to reach the working mixture saturation temperature to separate a part of the refrigerant fluid in the vapor phase. In the falling-film configuration, the working mixture is drizzled on a tube bundle or plates, while the heating fluid flows on the other side. According to Fujita [30], this configuration was more suitable than the nucleate pool-boiling because the thin-film enhances the desorption process. However, the wall temperature must be higher (around 12 °C) than the working mixture saturation temperature. When this condition is reached, the desorption process begins (by nucleation bubbles) at the solid-liquid interface [31]. In order to grow the bubbles and they may depart from the heating surface, a large heat transfer area at superheat thermal condition is required. Besides, falling-film desorbers show a “channeling” effect when the solution mass flow is low or the heat flux raised; therefore, useful heat and mass transfer area is reduced [32]. To enhance the heat and mass transfer processes, and to reduce the desorber size, various studies have been carried out in the desorber designs [33,34,35,36]. Additionally, the desorber efficiency is further enhanced by nanoparticles or surfactants, which improve the thermodynamic properties of the working mixtures [37,38,39,40] or by the use of ultrasonic waves [41,42].

An alternative to the conventional boiling desorption is the membrane separation processes. The desorption and desalination processes using membrane systems are similar in operation; the main differences are the salt type and the salt concentration in the feed stream. While the salt (NaCl) concentration in a typical membrane desalination process is around 4 wt% [43], the salt (LiBr) concentration in an absorption heat pump cycle is from 50 to 60 wt% [44]. Thus, different membrane technologies have been adapted to work with similar operating conditions than a conventional desorbers in an absorption heat pump cycle. In the following sections, membrane configurations and membrane devices proposed for absorption heat pump applications are presented and discussed in detail.

### 2.1. Membrane Distillation

Membrane distillation (MD) is a non-isothermal process used to remove a volatile chemical specie from a homogeneous liquid mixture; generally, an aqueous solution. The driving force is the partial pressure difference caused by a temperature difference between both sides of the membrane. The chemical specie with the highest partial pressure (most volatile component) evaporates at the membrane side, which is in contact with the liquid solution, then passes through the membrane pores, and later it condenses on a chilled surface, in the other membrane side. The hydrophobic nature of the membrane avoids the aqueous solutions penetrates the pores [45]. Figure 2 depicts the most uncomplicated MD process. The advantages of the MD regarding boiling desorption are: It has lower operating temperatures than the mixture saturation temperature; therefore, it is suitable for integrating low-grade or renewable thermal energy sources. It has relatively low operating pressures, the equipment can be much smaller, and it avoids corrosion problems by the use of polymeric materials [46]. Besides, the MD systems can be used in different configurations to improve the thermal efficiency of the separation process (e.g., integrating a regenerator to reduce the heat energy demand and increasing the system efficiency) [47]. For absorption heat pumps applications, desorbers based on MD process has been widely reported:

Riffat et al. [48] proposed an absorption refrigeration system which includes a pervaporation membrane desorber replacing the conventional boiling desorber. Figure 3 depicts the cycle described by the authors. There were two advantages of using a pervaporation membrane, i.e., components selection irrespective of working and reduced installation space. The membrane desorber was tested with the potassium formate/water (CHKO_2_-water) mixture and the potassium formate + cesium formate/water (CHKO_2_-CHCsO_2_-water) ternary mixture. The experimentation on the vapor absorption refrigeration (VAR) system, with the silicon membrane desorber, resulted in 10 °C of evaporator temperature at the source temperature of 80 °C with a constant condenser and absorber temperatures of 30 °C. The achieved COP was about 0.4.

Thorud et al. [49] experimentally studied the desorption process in a thin film membrane sandwiched with a porous aluminum plate. The study focused on the effects of the LiBr concentration, the height of the channel, and the pressure difference between the desorber and condenser in the desorption process. The membrane used was made of Polytetrafluoroethylene (PTFE). The author found that the refrigerant mass transfer via membrane was improved with the lower membrane thickness and higher-pressure difference. Moreover, the lower concentration also results in a higher refrigerant flow rate. The study also focused on the Reynolds number, Sherwood number, and Schmidt number to examine the heat flux, due to concentration difference. A drop was seen in Sherwood number at higher solution concentration; hence, it results in lower desorption rate of the refrigerant. The channel height improved the desorption rate, and the heat transfer resistance decreased as the channel height decreased. Kim et al. [50] proposed a small handy heat pump for electronic devices cooling. According to the authors, 100 W is the minimum dissipation heat in electronic devices, especially in microprocessors. This dissipated heat was used for cooling purposes by an absorption heat pump, which integrated membrane components. The microprocessor cooling was achieved by placing the microchannel merged with membrane for both condenser and desorber. With the microchannel membrane cooling component, 106 W of heat was dissipated out from the microprocessor. The heat inside the desorber was 20 W, and the achieved COP was 0.74. It was also reported that the evaporator pressure in the microchannel based membrane was higher than the existing heat achieving a higher COP.

Wang et al. [51] proposed a hollow fiber membrane desorber to minimize the size, as well as to reduce the heat energy supplied to a conventional boiling desorber. Membrane module made up of polyvinylidene fluoride (PVDF), and a working mixture with 50% of LiBr concentration was used. The desorption process was carried out at vacuum pressure condition, and two operating temperatures (83.5 and 88 °C) were evaluated. The impact of the feed flux was less for the vapor flow rate of the refrigerant compared to vacuum pressure and feed temperature. Based on the experimental results, the authors proposed an absorption system which includes a membrane desorber and demonstrated the feasibility and potential application of the desorption device. Figure 4 shows the system proposed by the authors.

Isfahani et al. [52] evaluated an absorption cooling system integrating membrane components for absorption and desorption processes with the water-LiBr mixture. The desorber’s size was 16.8 × 16.5 cm^2^, and microchannels with 160 μm thick, 1 mm wide and 38 mm long were machined on a Hastelloy plate. The authors analyzed the effect of the wall temperature, vapor and solution pressures, and the solution mass flow rate on the desorption rate. The desorption rate increased as a linear function of the wall temperature; however, around 100 °C, when the boiling regime started, the desorption rate increased exponentially. The authors found that the solution pressure and mass flow effects were negligible on the desorption process in the direct diffusion mode. The highest desorption rate was 0.0125 kg/m^2^·s at the boiling desorption regime. Authors concluded that the operating parameters in a membrane-based desorber could be optimized to increase the performance, unlike the conventional boiling desorbers. According to Bigham et al. [53], the molecular diffusion of refrigerant vapor in the membrane interphase is slow and to increase the processing speed, and the solution must be heated near to boiling temperature. However, the bubble formation inside the solution, at the inlet of the membrane channel, becomes a problem. Still, the process can be enhanced by a refill of the water-rich solution concentration continuously. The authors evaluated a membrane desorber with 16.8 × 16.5 cm^2^ size and with a microchannel size of 5.7 × 8.9 cm^2^ made up of Hastelloy, which was corrosion resistance. The method of refilling the water-rich solution had a desorption rate of 1.3 times higher than the boiling type vapor desorber and 1.7 times more than the permeable membrane type. Isfahani et al. [54] analyzed the physics of two different mechanisms (a) diffusion of water molecules in direct mode, and (b) based on the formation of water bubbles. The two mechanisms were analyzed based on the vapor and solution pressure parameters, wall temperature, and mass flux of the solution. The desorption rate by direct diffusion was affected by the solution concentration, vapor pressure, and wall temperature. On the other hand, the vapor pressure and wall temperature impacted in the second mechanism. The bubble formation increased the interphase area; thus, the vapor mass transfer was enhanced. However, bubble formation near the end of microchannels negatively affected the desorption process because some of these bubbles escaped with the solution stream. With the solution mass flow rate at 0.75 kg/h the total of bubbles formed were venting through the membrane, and with a solution flux of 3.25 kg/m^2^·s the bubbles exit rate increased up to 20% of the desorption rate. 

Hong et al. [55] proposed the LiBr-water absorption refrigeration system for automobiles. The vapor compression refrigeration (VCR) system is currently used in automobiles because of its compactness, but the VAR system can also be used in automobiles by replacing the normal absorption system by membrane absorption system. By replacing the hollow fiber generator in the place existing mechanical generator helps to minimize the size and shape of the VAR system. Moreover, the waste heat from the engine exhaust gas was used for the desorption process. A maximum cooling capacity of 2.88 kW was achieved at the generator temperature of 120 °C with a higher COP of 0.68. The COP of the membrane-based absorption system was reduced, due to a temperature drop in the membrane generator compared to the conventional absorption process, but the COP of the membrane absorption system was raised by the recirculation method. Venegas et al. [56] proposed a rectangular-shaped micro desorber for LiBr-water-based absorption chiller. The rectangular-shaped membrane technology was theoretically simulated to study the behavior in the LiBr-water absorption chiller. From the simulation, it was suggested to use smaller channels because the rate of cooling output was less compared to the rate of increase in the channel size. Hong et al. [57] analyzed the adiabatic heat and mass transfer processes in a membrane desorber. Some of the advantages in the studied membrane desorber were the improvement in the heat and mass transfer processes, thus reducing the size of the component and the reduction of the weight and the elimination of the corrosion, due to the use of non-metallic materials. The desorption rate behavior as a function of the working solution temperature and mass flux was studied. According to the authors, the desorption rate increases as the solution temperature and solution mass flux also increased. On the other hand, as the LiBr concentration decreases, the vapor mass transfer was improved. The highest desorption rate was 4.0 kg/m^2^·h at a solution temperature of 80 °C and mass flux of 240 kg/m^2^·s.

Venegas et al. [58] carried out a theoretical parametric study to analyze the effect of the membrane properties, design, and operating conditions on the desorption rate in a membrane microchannel desorber. According to the authors, the effect of the analyzed variables was higher in the boiling desorption mode than in the direct diffusion mode. The desorber’s size was affected by the solution concentration and the physical membrane properties in both desorption modes. During the desorber operation, the highest available thermal source must be selected, and, in the direct diffusion mode, the vapor pressure must be kept low. In another paper, Venegas et al. [59] evaluated a microchannel membrane-based desorber made up of stainless steel. Temperatures as low as 62 and 66 °C, with an inlet mass flow rate from 0.5 to 1.7 kg/h, were analyzed. The LiBr-water solution flowed through 50 microchannels in the rectangular shape desorber with a volume of around 104.6 cm^3^. Desorption rates of 0.0016 and 0.0042 kg/m^2^·s were achieved with cooling to desorber volume ratio of 415 kW/m^3^, and the achieved results were relatively higher compared to the small types of chillers.

The vacuum membrane distillation has been the most used configuration; in this configuration, the vapor condensation occurs in a separate condenser. Another configuration that has been studied is the air-gap membrane distillation (AGMD). In this configuration, the membrane and the cooling surface are separated by an air-gap, reducing heat loss by conduction through the membrane. However, the air-gap increases the vapor mass transfer resistance; thus, the permeate flux is lower than other configurations [60]. This configuration is more compact than vacuum membrane distillation because the desorption and condensation processes occur in a single component. To achieve the miniaturization goal of the absorption heat pumps, components with multi-process (e.g., desorber/condenser, evaporator/absorber, etc.) must replace the single-process components (e.g., desorber, condenser, etc.). Ibarra-Bahena et al. [61] adapted an AGMD module as a dual component (desorber and condenser). The authors carried out an experimental analysis to demonstrate the technical feasibility of this module as an alternative for replacing a conventional boiling desorber in an AHT system. The highest desorption rate reported was 2.4 kg/m^2^·h by using the Carrol-water mixture. With the studied configuration, the desorption process occurred at atmospheric pressure, so an energy source close to 100 °C was necessary for the proper evaporator operation in the AHT cycle. However, as the evaporator temperature increases, the revalorization temperature can be higher than an AHT system operated at vacuum pressure condition. Ibarra-Bahena et al. [62] evaluated a membrane desorber with the LiBr-water mixture. The desorption rate was from 0.30 to 9.69 kg/m^2^·h. The authors described a one-dimension heat and mass transfer model. According to the results, the LiBr concentration was the most restrictive parameter for the water vapor mass transfer at the membrane boundary layer; as the concentration of LiBr increased, the mass transfer resistance increased; however, as the solution temperature increased, the mass transfer process was improved. The membrane desorber based in the AGMD configuration can operate at atmospheric pressure conditions. This is an important advantage because the vacuum pump used to reduce the operating pressure in an absorption heat pump is not required.

An “ideal” ecofriendly absorption heat pump must be powered with renewable energies [63], and as it was discussed previously in this section, the desorption process by MD can use low-thermal energy sources, such as solar or geothermal energies. Some research studies had demonstrated this concept for absorption heat pump applications. Ibarra-Bahena et al. [64] analyzed a theoretical intermittent absorption chiller operated with the LiBr-water mixture, which integrated a membrane desorption process driven by thermal solar energy. In this configuration, the desorption-condensation and evaporation-absorption processes occur separately. The membrane desorber operated for four continuous hours, and after that period, the refrigerant fluid produced was 14.5, 11.6, and 7.2 kg per m^2^ of membrane area and with solution temperatures of 95.1, 85.2, and 75.1 °C, respectively. The solar system calculated included a 0.3 m^3^ thermal container and solar collector area of 30.2, 25.6, and 20.9 m^2^ for each operation temperature. With the experimental data from the membrane desorber, the authors simulated the absorption chiller system and concluded that the operating evaporator temperature was between 18 to 12 °C. In another paper, Ibarra-Bahena et al. [65] sized a solar collector system based on a membrane desorber’s performance. A total collection area of 37.4 m^2^ was calculated, considering the thermal requirements for a module with 1 m^2^ of membrane area. According to the authors, the membrane desorber driven by thermal solar energy can provide 16.8 kg/day of refrigerant fluid for the absorption chiller operation. Venegas et al. [66] proposed a solar-based absorption cooling system, which included a membrane desorber and the LiBr-water-nanofluids mixture. Three nanoparticles (CuO, Al_2_O_3_, and carbon nanotubes) were analyzed. The desorption rate increased 7.9% with the LiBr-water and carbon nanotubes mixture, respect to the LiBr-water solution base case. The required desorber modules were identified depending on different flow rates. The maximum cooling output was 645 W, with the highest number of desorber modules and a flow rate of 4 mg/s.

Table 1 summarizes some information of the literature reviewed on desorbers used for experimentations which include, membrane modules specifications as membrane area (*A_m_*), membrane thickness (*δ_m_*), pore size (*d_m_*), and porosity (*ϕ*), including also operational parameters, such as pressure potential (Δ*P*), temperature (*T*), and concentration (*X*).

By analyzing desorbers based on MD, in both theoretical and experimental studies, some of the optimum conditions were identified for successful implementation of them in the heat pumps. Usually, for conventional absorption heat pumps, the desorber performance is influenced by four variable parameters (solution temperature, concentration, mass flow and condenser pressure). In addition to this, the porosity, membrane thickness, and pressure difference play a major role when the membrane type desorber is replaced in the place of a mechanical compressor or a thermal compressor. From the literature reviewed, the following conclusions were obtained.
Higher pressure difference results in a higher desorption rate and reduces the wall superheat temperature; but the exit pressure was a constraint to condenser pressure. Hence, it was suggested to maintain a higher-pressure difference.Increasing the mean pore diameter increases the refrigerant mass flow rate, which increases the COP of the system, but it leads to weakening the mechanical strength of the membrane and could cause the membrane wetting. Hence, choosing and maintaining an optimum mean pore diameter is essential to get a high desorption rate without risk the membrane mechanical strength.Another important design condition in the membrane desorber is the membrane thickness, even though the membrane thickness gives mechanical support to the membrane, it increases the resistance to the desorption rate. Hence, lower membrane thickness with steel plate support is suggested.

Since the hydrophobic nature of the membranes used in MD desorbers, the working mixtures based on water as refrigerant fluid are the most used. However, the ammonia-water solution is an important mixture used for freezing applications in absorption heat pumps. Generally, in the conventional boiling desorption process, the main drawback is the trace of water in the ammonia vapor [67]. In order to solve this problem, the MD process could be an alternative. The ammonia vapor desorption, from a diluted aqueous solution, was analyzed in [68,69,70]. However, besides the conventional operation parameters (such as solution temperature, flow velocity, and ammonia concentration), the solution pH plays an essential role [71], which means some additives must be used to improve the mass transfer. Additionally, the effect of the high operating pressures required for the ammonia vapor desorption and condensation on the performance and mechanical resistance of the desorbers based on the MD process has not been studied yet.

### 2.2. Reverse Osmosis

In reverse osmosis (RO), a semipermeable membrane is used to separate a concentrated (in solute) solution from a pure solvent or a relatively diluted (in solute) solution. Since the semipermeable membrane is a selective barrier, only the solvent pass throughout the membrane pores freely. If the pressure exerted by the concentrated solution is higher than the osmotic pressure, the solvent flows from the concentrated solution to the diluted solution in the opposite way to the mass transfer by osmosis process [72]. Figure 5 shows a schematic diagram of the osmosis and RO processes.

The use of RO modules in absorption heat pumps has been studied in several reports: Su and Riffat [73] proposed and analyzed an absorption cooling system with an RO module as desorber. Figure 6 shows the proposed system. The operational parameters were calculated for the osmotic pressure required for different working solutions; however, these values were higher than the safe operating condition of the most used membrane modules, even the mechanical resistance could be at risk. For instance, LiBr-water mixture needs up to 6000 bars to carry out the desorption process, about 2000 bars for NH_3_-LiNO_3_ mixture, 700 bars for methanol/LiBr/ZnCl_2_, and around 350 bars for R134a/DMETEG mixture. According to the authors, the osmotic pressure is mainly affected by the refrigerant fluid thermodynamic properties. The main advantage of the RO process is to work at operating temperatures close to the ambient conditions; however, the operating osmotic pressure must be as low as possible in order to be used in absorption heat pumps.

Steiu et al. [74] carried out an experimental evaluation of a desorber based on the RO configuration, with the NaOH-water mixture. Based on the experimental results, the authors simulated the desorption process in-series arrangement and concluded that 99% of NaOH retention was reached with 2 RO stages for the assumed NaOH initial mass fraction of 0.039. The operating pressure was higher than 24 bar. The authors also analyzed the same RO system to separate NaOH from the ammonia-water-NaOH ternary mixture. The use of this mixture reduces the operating temperature of the absorption chillers, increasing the system COP. However, the absorption process was limited by the addition of NaOH; for this reason, a separation system must be included between the desorber and the absorber [75]. The analysis was made for three stages in order to reach 99% of NaOH retention and demonstrate the potential of RO systems in absorption cooling cycles. Rudiyanto et al. [76] carried out a thermodynamic analysis for an RO membrane system where it was used as desorber with LiBr-water mixture. According to the study, the desorption rate and permeate quality were affected by the operating pressure. On the other hand, the working mixture flow rate also impacts the temperature difference between the inlet and outlet stream. The calculated irreversibility of the RO desorber increased only 0.06% when the operating pressure increased from 440 to 520 kPa, this was because the environmental temperature condition was considered for the desorber operation. The calculated COP for the absorption refrigeration system operated by the RO membrane desorber was in the range of 1.15 to 1.17.

### 2.3. Electrodialysis

Electrodialysis (ED) is a separation process in which electrically charged membranes are employed to remove ionic species from a liquid solution (generally an aqueous solution) and other uncharged components, and the driving force used here is the electrical potential difference [77]. Figure 7 illustrates the principle of electrodialysis.

Since the absorption heat pumps working mixtures were integrated by ions aqueous solutions and the research on this technology is growing up because this membrane configuration is being interesting and alternative to the desorption process. Li and Zhang [78] proposed and analyzed a membrane regenerator driven by electric power generated by renewable sources. This regenerator operated as an ED stack, which consisted of several cells in-parallel arrangement between two electrodes. The aqueous solution is concentrated and “desalinated”, respectively, in interspersed cells. The system is shown in Figure 8. The ED module replaces a conventional desorber and a condenser. The proposed system operates as follow: Initially, an aqueous solution stream is pumped from the storage tank 1 to the dilute cells (valve 2 is open, while valves 1, 3, and 4 are closed); meanwhile, the diluted LiBr solution (pumped from the absorber) is supplied to the concentrated cells. The working mixture is concentrated in each step of the regeneration process. A part of the concentrated solution is accumulated inside the storage tank 2 (valve 7 is open, while valves 5 and 9 are closed). The absorbent concentration in the dilute cells feeds stream decreases with the regeneration process. This diluted solution is pumped continuously (in a loop) from the dilute cells to the storage tank 1 (valve 6 is open, and valve 8 is closed). This cycle continues until the concentration of the diluted stream is close to zero; this condition means only pure water is present in this stream; thus, the desorption process ends. After that, the concentrated solution inside the storage tank 2 is pumped to the dilute cells (valve 4 is open, and valve 2 is closed). The refrigerant fluid in the tank 2 is sent to the water storage tank (valve 1 is open, and valve 3 stays closed) and pumped to the evaporator (in order to produce the cooling effect), and the concentrated solution is lead to the absorber from the storage tank 2 (valve 9 is open, and valve 5 is closed). Finally, the storage tank 1 replaces the storage tank 2 before starts the cycle again (valves 5 and 8 are open, while valves 7 and 6 are closed).

Theoretical analyses were carried out for an air-conditioning absorption system, which included the ED regenerator, and conclude that the COP of the proposed system was similar to the conventional VCR system. However, to improve the COP, the concentration of the regenerated absorbent solution and the applied voltage should be as low as possible. Additionally, the proposed system also showed the flexibility of electric power supply because it can use renewable energy sources assisted with energy storage devices. Li et al. [79] analyzed the effects on COP, cost-effectiveness, and economy of the ED regeneration system with the LiBr+CaCl_2_-H_2_O mixture. According to the authors, the COP increases as the concentration of the total solutes decreases; however, with a constant total solutes concentration, a higher concentration ratio of LiBr increases the COP. On the other hand, with a higher concentration ratio of CaCl_2_, the cost-effectiveness increases because the CaCl_2_ is cheaper than the LiBr, but a high CaCl_2_ concentration limits the mass transfer process. After analyzing a wide range of the concentration ratios, the authors concluded that the most suitable concentration of the total solutes was between 50 and 58%, in terms of the cost-effectiveness value. In order to corroborate the theoretical results reported previously in [78], Li et al. [80] carried out an experimental evaluation of an ED regeneration module with the LiCl-water mixture. The regenerator size was 350 mm height, 200 mm width and 123.5 mm depth, and included 25 membrane pairs with a total membrane area of 0.021 m^2^. The current utilization efficiency, defined as a measure of how effectively the ions pass through the membranes at a constant applied current, was from 30% to 50%. Moreover, this parameter decreases as the solute concentration increases. Besides, the current utilization efficiency increased as the current intensity decreased because of the energy losses also decreased. The COP of the system was around 3 at 5 A of current intensity. On the other hand, the COP was lower than 1 when the current intensities were 10 A and 15 A, respectively. According to the study carried out by Sun et al. [81], a higher concentration of the absorbent salt leads to a lower membrane permselectivity, reduces the current efficiency, limits the water mass transfer rate, increases the electrical resistance and the energy losses increase by the Joule effect. Thus, the ED regenerator system performance was unstable when the concentration difference was increased [82]. In order to solve this drawback, Ding et al. [83,84] proposed and analyzed a multi-stage ED regeneration system. An ED module with the same size reported in [80], and the working solution of the LiCl-water mixture was used. The experimental COP of a double-stage system was around 3, which was 15% higher than the single-stage system. Finally, it was concluded that more regenerators reduced the voltage ratio, which increases the current efficiency; thus, the COP increases with the increase in stages. However, an increase in the number of regeneration stages leads to the system turn into complex and expensive. Liang et al. [85] analyzed an absorption cycle which integrated an ED desorber, placed between a conventional boiling desorber and the absorber. The NH_3_-H_2_O-LiBr mixture was considered for the theoretical study. The key function of the ED device was to separate the LiBr from the solution stream going to the absorber. For the simulation, a separation efficiency of 90% and a desorption temperature of 120 °C were assumed. The maximum calculated COP with the proposed cycle was 0.71, which was 26.9% higher than the conventional NH_3_-H_2_O cycle, and 12.6% higher than the cycle with the ternary mixture, but without the ED desorber. Other salts, such as NaOH and LiNO_3_, can be used as additives with the NH_3_-H_2_O mixture, and an ED desorber could be an interesting option to improve the performance of the absorption cooling cycle [86].

## 3. Membrane-Based Absorbers

The absorber is one of the key components in absorption heat pumps used either for cooling or heating. This component is used to absorb the refrigerant in the vapor phase coming from the evaporator by means of the solution coming from the desorber. During this process, a large amount of heat is rejected [87,88,89]. Moreover, it depends on the mixture of refrigerant and absorbent used, such as ammonia-water, LiBr-water, NaSCN-ammonia, R134a-DMAC, etc. The process is similar to all kinds of absorption systems like heat pumps, chiller, cogeneration, etc. [90,91,92,93]. Due to the large amount of heat rejected, this component is usually very big and costly compared to the other components of the system. Also, higher exergy destruction has been detected [94,95,96,97]. To increase the heat rejection and mass transfer between vapor and liquid, and to improve the absorber efficiency, diverse types of absorbers have been designed and proposed for the absorption cooling systems. Among other things, the designs include changes in the flow’s direction (upward and downward and falling film type), the use of magnetic effects to improve the mass transfer, or to incorporate additional fins increasing wettability [98,99,100]. Even though the absorber efficiency has been improved by diverse modifications in design, there is still a huge amount of heat transfer that makes the heat exchanger in a bigger size. The importance of the absorber cannot be neglected because it directly influences the cooling and heating output, and hence, it directly affects the system performance. To overcome the existing challenges, the use membrane has been proposed in the absorber to do similar work with reduced size and shape. Membrane contactors provide the interphase between vapor and liquid required for the absorption process. The absorber impacts are directly dependent on the parameters of refrigerant, absorbent mixture ratio, and water temperature used to cool the absorber [101,102,103,104]. The similar parameters and other parametric conditions used and influenced in the membrane type absorber have been studied in detail.

Yu et al. [105] proposed and suggested to use the osmotic pressure in the field of refrigeration. The evaporator, generator, expansion valve, and condenser are similar as in conventional absorption heat pumps, but instead of an absorber, a cellulose triacetate membrane was used. This membrane allows only the pass of water in the vapor phase. It was reported that during the operation of the absorption system at a constant temperature, the nature of source temperature was dynamic, and also it was concluded that the system could not act as a heat pump. Sudoh et al. [106] studied the properties of LiBr-water solution through the penetration of the permeable membrane. Hot and cold junctions were kept two sides of the permeable membrane and maintained the cold junction temperature of 15 °C. The hot junction temperature was varied from 35–100 °C, and the consent temperature was kept constant with the help of hot water by the continuous starring process. The experimentation was carried out by using the LiBr-water solution concentration in a range from 0 to 55%. The Analysis results in that the refrigerant vapor penetration through the membrane was reduced when the LiBr-water solution concentrations were increased, as well as a decrease in solution temperature. A study on the properties and characteristics of the fiber membrane used in a LiBr-water VAR chiller was carried out by Ali and Schwerdt [107]. The factor which affects the performance of the membrane VAR chiller was also studied based on the wettability, permeability, inlet pressure, and condensation of refrigerant vapor. It was proposed to use a porous capacity of up to 80% with a pore size of 0.45 µm and a thickness of 60 µm. It was advised to use a metal layer to support the thin membrane to provide mechanical support. The characteristic should be maintained while designing the membrane cooling system, i.e., the condensation of vapor should not take place in the membrane, maintaining the small pore size and high inlet pressure. The experimentation was carried out with the LiBr-water concentration of 50.8 to 54% by varying the evaporator temperature in the range of 16–25 °C. The rate of water vapor was increased for the reduced thickness of the membrane layer and an increase in mean pore diameter when the solution temperature was varied from 24–29 °C. Ali [108] analyzed by numerical simulation, a multilayer membrane desorber using the plate-and-frame configuration. The individual flowing channels for the LiBr solution, cooling water, and the water vapor, were formed by the membrane sheets and the heat transfer plates in a parallel-arrangement. The hot junction temperature of 85 °C and cold junction temperature of 4 °C with an environmental temperature of 25 °C were used for the source conditions for mathematical modeling. The temperature differences among the different layers were caused because the membrane-side heat transfer resistance was the most influential factor in the overall heat transfer coefficient value. The solution flow rate was increased, due to the upward flow direction of refrigerant vapor and the flow of refrigerant vapor may be zero at the top of the absorber. The ratio of the membrane area to the heat transfer area was changed from 2.26 to 1.27 when the solution channel thickness changed from 1 to 2.5 mm.

Woods et al. [109] designed and experimentally analyzed a membrane absorption-type heat pump. The proposed system included a membrane absorber using two hollow-fiber rows; the salt solution flowed inside one, while the water flowed in the other row. The membrane fibers were made of Oxyphan^®^. An air-gap was placed between both rows to increase the temperature lift, defined as the temperature difference between the outlet solution stream and the inlet water stream. The LiCl_2_ and LiCl with water mixtures were analyzed experimentally, even though the cost of the LiCl_2_ was 20 times higher than LiCl, and a higher temperature lift was seen in LiCl. The LiCl-water solution concentration of 39% was used for experimentation. The temperature lift was up to 9 °C. Besides, based on the mathematical model, the temperature lift could be up to 14 °C with higher-porosity and larger pore-size membranes. Still, the temperature lift can be raised up to 14 °C by maintaining the high porosity and high pore diameter. The authors simulated and compared the performances of a hollow-fiber membrane module with membranes made with Oxyphan^®^ and Accurel^®^, assumed the same membrane physical characteristics, and considered a tortuosity value of 2. According to the authors’ results, the mass transfer coefficient with the first type of membrane was 4.2 times higher than the second type. The water vapor mass flux raised 2.1 times with the Accurel^®^ membranes respect to the Oxyphan^®^; however, the air-gap increased the mass transfer resistance.

Isfahani et al. [52] investigated the implementation of the nanofiber membrane in a VAR system. The nanofiber membrane was individually replaced in both absorber and desorber. Based on the permeability study, a membrane with a pore size of 1 µm was used for the absorber, whereas the pore size of 0.45 µm was used for the desorber. The analyses were separately carried out for both absorber and desorber. The parameters of cooling temperature, vapor pressure were used to study the absorption rate, whereas the solution and vapor pressure, velocity, and wall temperature were used to study the desorption process. The absorption rate was increased with the increase of the pressure potential and the solution velocity. A 50% increase in the flow rate of refrigerant was observed in boiling mode. Figure 9 illustrates the absorption process described by the authors.

Isfahani and Moghaddam [110] experimentally studied a superhydrophobic membrane absorber with constrained microchannels in a VAR system with the LiBr-water mixture. The solution film thickness, cooling temperature, vapor pressure, and solution flow velocity were analyzed. The desorption rate was 0.006 kg/m^2^·s for the solution velocity of 5 mm/s and channel thickness of 0.1 mm. An increase in solution velocity and a decrease in solution film thickness helped to boost the mass transfer process. The absorption rate with the membrane absorber was 2.5 times greater than the falling film absorption configuration. According to the authors, the reduced size of the membrane absorbers could help develop new absorption cooling systems for small scale applications. Isfahani et al. [111] studied the performance of a membrane absorber with microstructures on the solution flow channel to promote vortex generation and to improve the mass transfer process. The microchannels were 0.5 mm deep, 1 mm wide, and 195 mm long, while the microstructures were 0.2 mm deep and were in a 30° herringbone arrangement. The authors reported that high absorption rates were achieved with low pressure-drops in the solution channel by the principle of micromixing generated by the microstructures. The results also showed that the absorption rate was higher in a 0.5 mm thick solution channel than in a 0.1 mm thickness solution channel without microstructures. Hence, the authors demonstrated that it is feasible to build a large-scale absorption system using this principle.

Asfand et al. [112,113] reported numerical simulations to analyze the effect of the membrane characteristics (pore size, porosity, and thickness) and the operating conditions on the absorption process in a plate-and-frame membrane desorber with the LiBr-water mixture. The analyses demonstrated that the solution channel thickness was the most restrictive parameter for the mass transfer process. The absorption rate increased three times when the solution channel thickness was reduced from 2 mm to 0.5 mm. Also, a 0.005 m/s flow velocity of the solution was suggested to improve the absorption rate. The simulation study showed that a solution channel length between 100 and 200 mm was best suitable for the membrane desorber. Besides, the vapor mass transfer resistance through the membrane decreased as the pore diameter and porosity increased; however, as these parameters increased, the membrane mechanical strength was reduced. This problem was also reported by Isfahani et al. [52] when the membrane pore diameter was higher than 1 μm. An opposite behavior was observed with the membrane thickness, as this parameter increased, the mass transfer resistance increased, and the membrane mechanical strength increased. The absorption rate increased with the following operating conditions: High solution mass flow rate, high vapor pressure, and high inlet LiBr concentration. Another numerical studies [114,115] were carried out to analyze the geometric features and the operating conditions in a membrane desorber integrated by microchannels. The aqueous LiBr solution was considered. The effect of the vapor pressure on the absorption process was higher compared to other operating variable parameters. The channel width of the working solution and the cooling water, wall thermal conductivity, vapor superheating, and flow rate of cooling were not much disturbance for the membrane absorption refrigeration system compared to porosity, solution temperature, and vapor pressure in non-adiabatic operation mode. In the adiabatic operation mode, cooling water was not required, which is one of the most relevant advantages, since avoiding the used cooling towers; however, smaller channel lengths were suggested. The LiBr-H_2_O, LiNO_3_-NH_3_, and LiCl-H_2_O mixtures were analyzed with the same absorber configuration [116]. A membrane absorber with 13 microchannels of 0.15 mm height, 1.5 mm width, and 50 mm length was simulated. A membrane with 60 μm thickness, 80% porosity, and a pore diameter of 1 μm was considered. The analysis results showed that the LiNO_3_-NH_3_ mixture had a high cooling effect on the absorption volume ratio compared to the other two working mixtures. Additionally, the desorption temperature was lower than with the LiBr-H_2_O mixture. The lowest desorption temperature was achieved with the LiCl-H_2_O mixture; however, the COP was the lowest. The simulated membrane desorber was small in size compared to the conventional falling film absorbers for similar cooling capacities.

Huang [117] proposed and analyzed a parallel plate quasi-counter flow membrane in an absorption heat pump by extracting the heat and reusing it for fluid heating applications. The device, described by the author was similar to a membrane system used for the air dehumidification process by a liquid desiccant system, in which the refrigerant fluid and the absorbent are separated by a porous membrane which allows that only the refrigerant fluid vapor passes through it. However, the author analyzed the heat transfer process when an air-gap was placed between the water and the solution channels. The author also tested modified PVDF membranes with surfaces coated by silica gel. Based on a mathematical analysis, the author made a conclusion that the solution temperature lift (defined as the increment of the thermal level of the solution by the absorption process) and efficiency was increased by 9.1% compared to those of a cross-flow one. Huang [118] also proposed and evaluated the heat and mass transfer in a membrane process in the absorption heat pump. The composite membranes were made up of a layer of a combination of PVDF and PVA (polyvinyl alcohol). Layers of the membrane were framed and arranged in a cross-flow direction with an air-gap between the membranes to increase more absorption rate. The fiber diameter and packing fraction were optimized theoretically as 1.2 mm and 0.48, respectively. Higher solution temperature was required for hollow membrane compared to the permeable membrane, but the pressure drop in the hollow fiber membrane was less. An increase in packing fraction resulted in lower pressure, and also the skin layer was more dominant than the porous layer.

The replacement of membrane in the absorption and desorption for the ammonia-water mixture VAR system was also studied theoretically and experimentally by Berdasco et al. [119]. The PTFE membrane sheet laminated with polypropylene layer support, looking like the sandwich model was used for the experimentations. The achieved highest absorption rate with the ammonia-water mixture was 4.7 × 10^−3^ kg/m^2^·s for 45 kg/h of solution flow rate and working pressure of 1.3 bar. The best suitable membrane diameter was between 0.03 and 0.10 µm for the ammonia-water mixture. Higher than the 0.10 µm lead to the passage of the absorbent solution on the other side. In another paper, Berdasco et al. [120] studied the absorption process in a hollow fiber membrane module with the ammonia-water mixture. Usually, the hollow membrane cooling system will on the basis of the absorption and desorption process, whereas the proposed system will work on the adiabatic process. The study focused on the parameters of ammonia concentration, solution temperature, and absorption rate and gas temperature in both axial and in the radial direction of the flow process. The cooling water was not required during the absorption process. García-Hernando et al. [121] evaluated three types of flat membranes with different pore diameters and membrane thicknesses in a microchannel absorber operating adiabatically with the LiBr-water mixture. The absorber configuration was plate-and-frame, the working mixture flowed inside the microchannels, and a hydrophobic membrane separated the circulation microchannels and the vapor of the refrigerant fluid. The absorption rate value was from 0.0015 to 0.0026 kg/m^2^·s, and it increased linearly respect to the solution mass flow rate. The mass flow rate increased two times as the pore diameter decreased from 1 to 0.45 μm to achieve similar absorption rates. On the other hand, as the difference in thickness increased from 25–51 to 76–127 μm, with a constant mass flow rate, it needed two times the pressure potential for similar absorption rates. The authors concluded that the advantage of this membrane absorber is the modularity; it means that the absorption rate could be scaled for specific operating conditions. De Vega et al. [122] analyzed the performance of a microchannel membrane absorber operating non-adiabatically and adiabatically, with the LiBr-water mixture. In the first mode, the absorption heat load was removed by a cooling fluid circuit, and in the second mode, the absorber operated without cooling. The absorption rate in the first mode was 0.005 kg/m^2^·s; meanwhile, in the second mode was 0.003 kg/m^2^·s. The cooling capacity ratio was 600 and 400 kW/m^3^ for each operating mode, respectively. Although the value was lower in adiabatic operation, a cooling system used to remove the heat from the absorption process was unnecessary; thus, the absorption chillers could be more compact and easier to operate. Table 2 summarizes the literature of experimental studies using membranes in absorbers.

One of the most important differences between the vapor compression and absorption heats pumps were the component or principle absorption. The study on the feasibility of the membrane devices in the absorptions process discussed in detail and the conditions were given below.
The theoretical and experimental studies resulted that the pore size should be as small as possible to get high permeability; experimentally, 0.45 to 1.0 µm of pore size was suggested for the porosity level of 80%. The leakage of liquid will happen at the pore diameter higher than 0.10 µm for the ammonia absorption system.The pore will be blocked by condensation in the pores; hence, high inlet pressure should be maintained, and capillary condensation should be avoided. The maximum membrane thickness of 60 µm was suggested, and it can increase with mechanical support.The rate of absorption was increased with a difference in the pressure between the water vapor and solution flow rate. Hence, a higher-pressure difference can be maintained without affecting the evaporator temperature.The adiabatic operation can reduce some auxiliary components (such as a cooling tower, water pumps and pipe) in an absorption cooling system.

## 4. Membrane-Based Heat and Mass Exchangers

In order to improve the overall performance of absorption heat pumps, a solution heat exchanger (also known as economizer) is placed between the absorber and generator. The key function of this component is to recover sensible heat from the hot solution stream; this way, the generator heat load and the absorber heat rejection are reduced [123]. By using the solution heat exchanger, the COP can be improved up to 60% [124]; therefore, this is an essential component in absorption heat pumps. However, the conventional solution heat exchangers are usually heavy and costly because they are made with metallic materials (such as stainless steel, copper, etc.). Membrane modules can be used to heat recovery (like a conventional heat exchanger) but, besides this, they can “recover” a part of the mass of the refrigerant fluid from the hot and concentrated solution stream (from the desorber) to the cold and diluted solution stream (from the absorber). The temperature difference between both solution streams provides the driving force for heat and mass transfer processes. With the mass recovery, the absorption process is enhanced because the mass transfer coefficient increases as the absorbent concentration increases, and the temperature decreases in the solution stream entering the absorber [125]. Thus, membrane devices can improve the absorption heat pumps performance. The advantages of membrane heat and mass exchangers regarding conventional-based-metallic devices include cost reduction, high corrosion resistance, simplicity of manufacturing and maintenance, and a large surface area/volume ratio improving the heat transfer process [126,127]. The membrane-based heat and mass exchangers have been proposed and studied by diverse authors. Chen et al. [128] mathematically analyzed a membrane module operating as a heat exchanger and an absorber simultaneously with the ammonia-water mixture. According to the authors, due to the membrane module operated as a dual component, the volume can be reduced by 69%, and the useful heat and mass transfer area increased 4.3 times regarding conventional falling film absorbers, at a similar operating condition. Also, the exergy loss can be reduced by 26.7%, and the COP of the system increased by 14.8%. Woods et al. [129] theoretically modeled a membrane heat pump with CaCl_2_-water mixture. The refrigerant fluid and solution were separated by a vapor-permeable membrane which helped to lift the solution temperature. According to the authors’ results, with two membranes separated by an air-gap lifted the solution temperature up to 185%. It also resulted that the air-gap made the temperature lifts of 6, 16, and 24 °C for the inlet temperature of 15, 35, and 55 °C, respectively. It was suggested to use high porosity, low thermal conductivity, and thin membrane to improve the performance of the system. These membrane characteristics helped to achieve the higher COP. Wang et al. [130] reported a mathematical model to solve the heat and mass transfer processes in a 10 kW of cooling capacity-membrane heat exchanger operated with the LiBr-water mixture. A hollow membrane module, with an outer diameter of 50 mm, 800 mm length, an effective 1.2 m^2^ of membrane area, and a bundle with 600 individual hollow fiber membranes, was simulated. PVDF hollow fiber membranes with 0.16 μm of pore diameters, 85% porosity, and 150 μm thickness were considered. The operating conditions analyzed were: A solution with a 55% of LiBr concentration, 99.8 °C of temperature, and 0.0426 kg/s mass flow rate inside the hollow membrane fiber bundle, and a solution with 50% of LiBr concentration, 40 °C of temperature and 0.0469 kg/s mass flow rate in the shell side, in counter flow arrangement. From the simulation it was found that the LiBr concentration increased from 55 to 55.5% and the temperature decreased from 99.8 to 55 °C in the concentrated stream; while the LiBr concentration decreased from 50 to 49.5% and the temperature increased from 40 to 80 °C in the diluted stream. In a similar way that in conventional shell and tube heat exchangers, the counter-flow process has high efficiency, due to the high mean temperature difference.

A membrane-type heat and mass exchanger was proposed by Hong et al. [131] which was made up of hydrophobic hollow fiber membranes and it was tested with concentrated and diluted solutions of LiBr. The authors carried out a theoretical analysis; the concentrated and diluted solution temperatures of 80 and 40 °C, respectively, were assumed, and two different modes were studied, vacuum and direct mode, with a velocities range from 14 to 55 cm/s at fixed concentrations of 56% and 54% for the concentrated and diluted solutions, respectively. The mass recovery in the direct contact mode was disturbed by the heat conduction of the layer. The mass recovery in the vacuum mode was improved by increasing the temperature difference between the hot and cold solution streams. In Figure 10, a schematic diagram of the proposed VAR system, which integrated two membrane modules, is described. To improve the mathematical model of the heat and mass membrane-exchangers with the direct contact membrane distillation configuration, the concentration polarization effect in the boundary layer and non-constant mass transfer coefficient (by the effects of the local temperature and concentration) must be taken into account [132].

Some conclusions arising from the reported studies are the following:Direct contact membrane distillation configuration could replace the conventional SHE in an absorption cooling system; however, the mass transfer is improved by the vacuum membrane distillation.Membranes with high porosity and low thermal conductivity improves the heat and mass transfer rates in a single-stage membrane heat exchanger.Membrane-based heat and mass exchangers can operate as dual components (SHE-absorber); thus, reducing the volume and size of the absorption cooling systems.The membrane-based heat and mass exchangers can operate with working mixtures based on water and ammonia as refrigerant fluids.

## 5. Innovation in Heat Pumps with Membrane Processes

The advantages of membrane processes have promoted several new technical developments in absorption cooling systems. A breakthrough in the membrane-based heat pump and novel design outcome and its related patents are given in Table 3. The studies on the absorption-type heat pump were very high compared to the study on compression-based heat pumps. However, in the patents, 65% of patents were issued to compressed type heat pumps, whereas it was only 19% for the absorption type membrane heat pumps; hence, more innovative concepts and design related to absorption-type membrane heat pumps should be focused to make membrane-type heat pump as marketable. However, many innovations have come out into the market by the use of membranes in the field of heat pumps. There are still many opportunities to do new studies and developments using membranes, not only in absorption systems, but also in more complex systems for the simultaneous production of power and cooling [89,92], and cooling and desalination systems [133], among others.

## 6. Conclusions

A detailed study and role of the use of different membrane configurations in the heat pumps were studied, and the review focused on individual components, such as desorber, absorber, and heat and mass exchanger. Based on the literature review, the following conclusions were found:DesorberThe most widely used membrane configuration in the heat pump is membrane distillation (MD) in that flat sheet, and hollow fiber configurations are the most broadly used modules.The desorption process by MD configuration occurs at lower temperatures than the boiling point of the working mixtures and at atmospheric or vacuum pressure conditions.Since the membrane in the MD is hydrophobic in nature, this configuration is suitable for liquid mixtures based on water as refrigerant fluid.The MD-based desorbers provide compact configurations, such as the AGMD, in which the desorption and condensation processes occur in a single component.The advantages of the MD modules are lower operating temperature and pressure, high mass transfer area, modularity, and scalability. Some identified problems are membrane fouling, the durability of the membrane, and membrane materials.The desorption process, in the RO membrane configuration, can be carried out at operation conditions equal to temperature as low as the ambient condition. However, the osmotic pressure is too high than the safe operating limits of the commercial membrane modules.An absorption heat pump that integrates an RO-based desorber could operate without a condenser.Nonetheless, combining RO membrane modules and conventional boiling desorbers can improve absorption cooling cycles because new working mixtures with additives will enhance the mass transfer process.Desorption by ED modules is a new attractive alternative for absorption cooling systems because the operating temperature is as low as the ambient condition.The ED modules operated absorption cooling system uses high-grade electric energy; it can be supplied by renewable sources, such as solar, wind, hydraulic, and others.The refrigerant fluid separation by ED process is carried out in the liquid phase, thus a condenser is not required.However, a multi-stage ED system is proposed in order to improve the desorption process, but an increase in the number of regeneration stages will turn the system turn into complex and expensive.AbsorberThe theoretical and experimental studies resulted that the pore size should be as small as possible to get high permeability; experimentally, 0.45 to 1.0 µm of pore size was suggested for the porosity level of 80%. The leakage of liquid will happen at the pore diameter higher than 0.10 µm for an ammonia absorption system.The pore will be blocked by condensation in the pores; hence, high inlet pressure should be maintained, and capillary condensation should be avoided. The maximum membrane thickness of 60 µm was suggested, and it can increase with mechanical support.The absorption rate increases as the pressure potential, and the solution flow rate increase. Hence, a higher-pressure difference can be maintained without affecting the evaporator temperature.Microchannels used in some membrane absorber design provide high heat transfer coefficients; thus, the mass transfer process is improved. Besides, induce vortex formation inside the solution channel by microstructures which lead to high absorption rates.The adiabatic operation can reduce some auxiliary components (such as a cooling tower, water pumps and pipe); thus, the absorption cooling systems could be used for small scale applications.Heat and mass exchangerHigh porosity and low thermal conductivity of the membrane results in a high heat transfer rate in a single stage membrane heat exchanger.The use of the membrane-based heat and mass exchangers improve the COP, since heat and mass transfer simultaneous processes occur instead of only the heat transfer process.Membrane-based heat and mass exchangers can operate as hybrid components (SHE-absorber). It leads to a volume and size reduction of the absorption cooling systems.The membrane-based heat and mass exchangers can operate with the most used working mixtures.

The membrane devices provide an opportunity to develop more compactness and energy-efficient absorption heat pump systems, and several patents have been developed. However, until now, the membrane technology is at the research level or laboratory scale. The replacement of absorber, desorber, and heat and mass exchanger in the absorption heat pump systems is under progress, and a commercial level has not been reached. Besides, the experimental and theoretical investigations have been focused on demonstrating the technical feasibility of these devices and analyzing the heat and mass transfer mechanisms; but, techno-economical and exergoeconomic analyses are necessary to compare the advantages and drawbacks of the membrane devices respect to the conventional components. Additionally, the research about new materials must be increased to develop membranes for this specific application to avoid fouling, wetting, and mechanical failures.

## Figures and Tables

**Figure 1 membranes-10-00216-f001:**
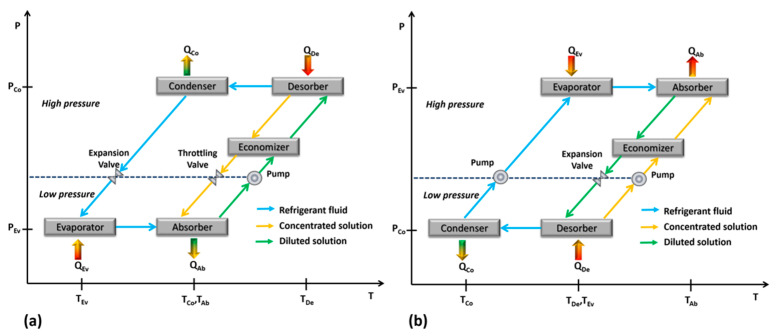
An absorption chiller diagram (**a**), and an absorption heat transformer diagram (**b**).

**Figure 2 membranes-10-00216-f002:**
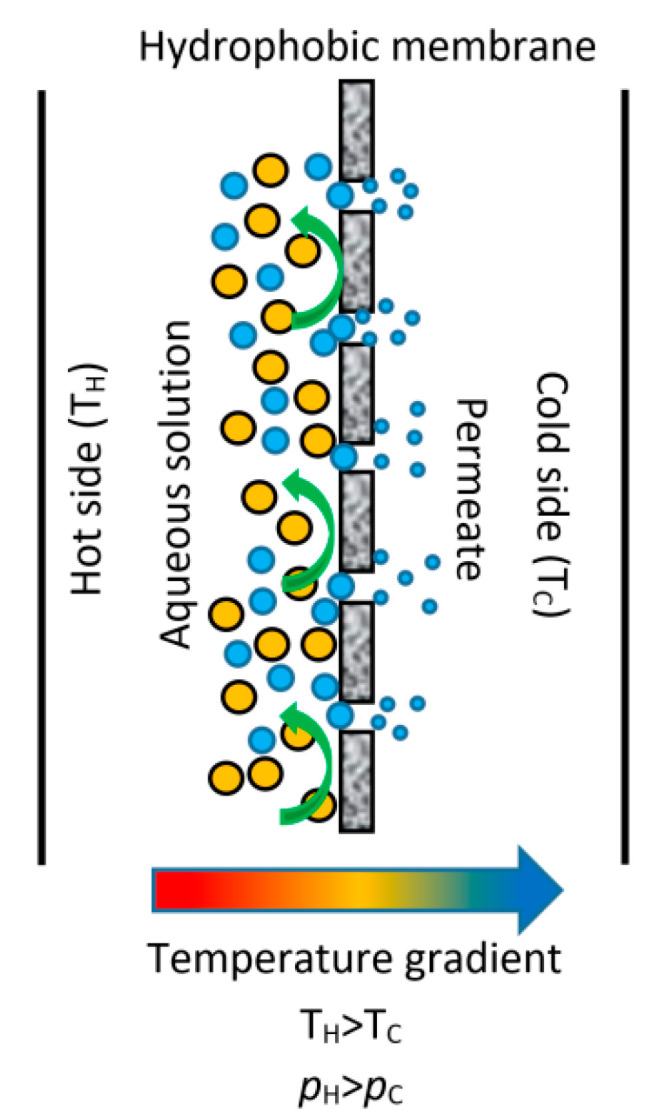
Basic membrane distillation (MD) configuration.

**Figure 3 membranes-10-00216-f003:**
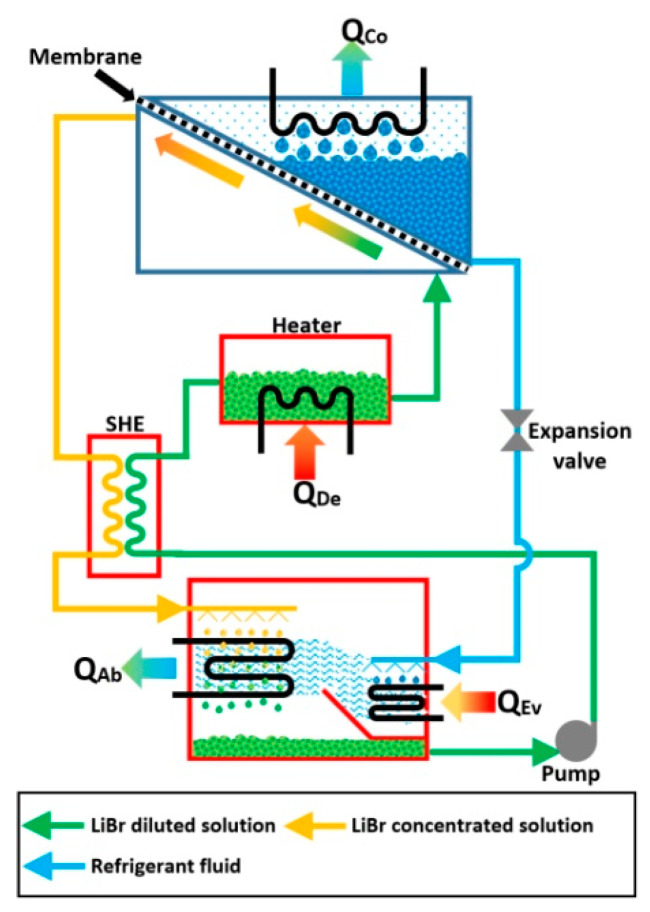
Membrane absorption system described by Riffat et al. [48].

**Figure 4 membranes-10-00216-f004:**
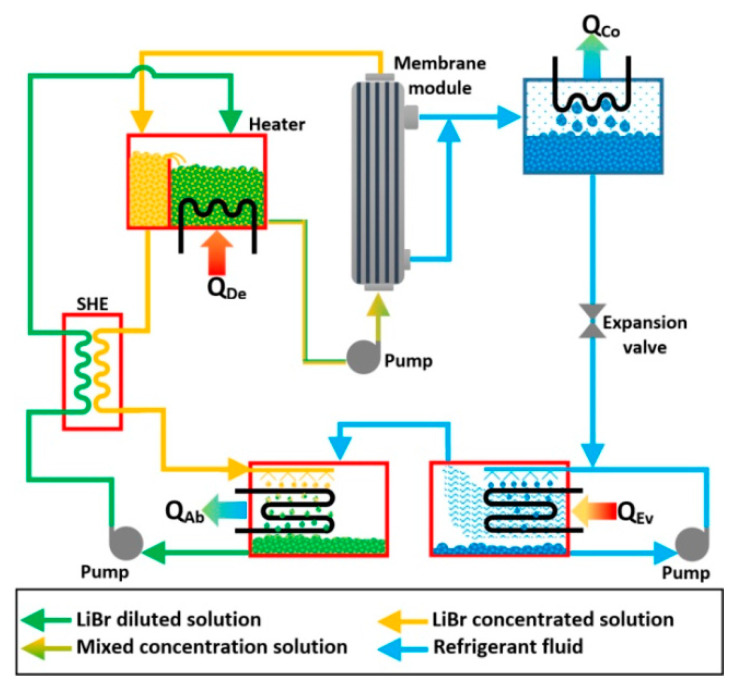
Membrane absorption system proposed by Wang et al. [51].

**Figure 5 membranes-10-00216-f005:**
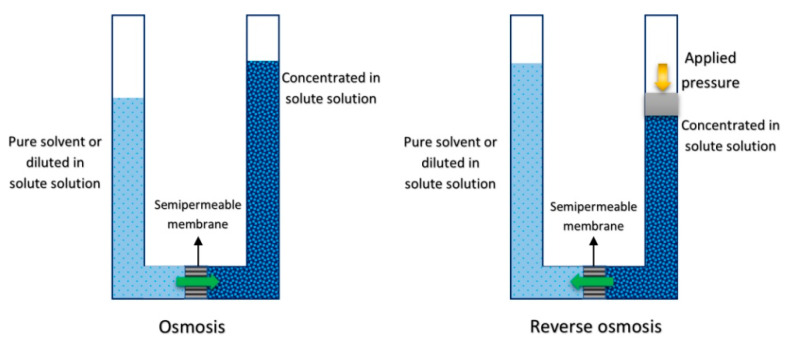
Schematic diagram of osmosis (**left**) and RO (**right**) processes.

**Figure 6 membranes-10-00216-f006:**
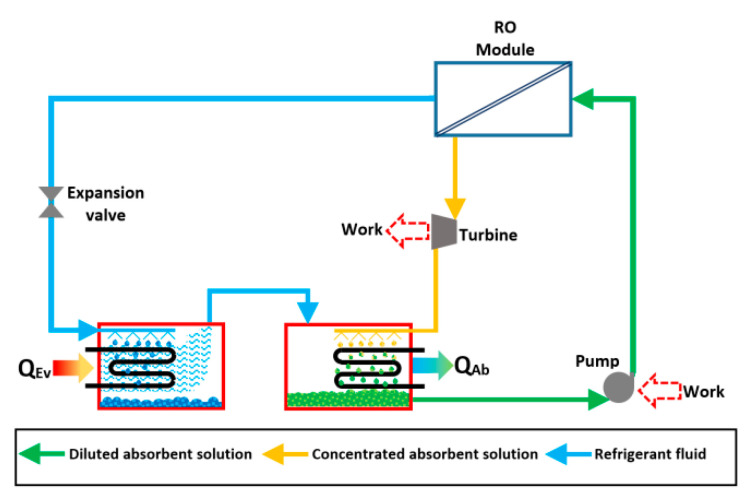
Schematic diagram of the cycle proposed by Su and Riffat [73].

**Figure 7 membranes-10-00216-f007:**
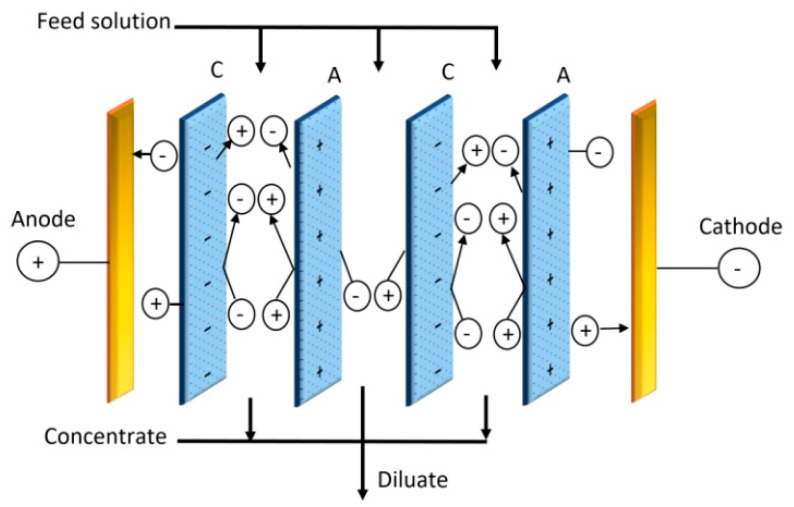
Schematic diagram of the electrodialysis (ED) process.

**Figure 8 membranes-10-00216-f008:**
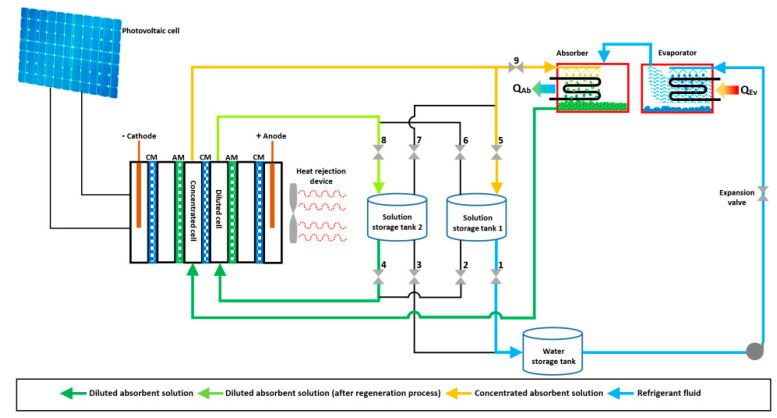
Schematic diagram of the system proposed by Li and Zhang [78].

**Figure 9 membranes-10-00216-f009:**
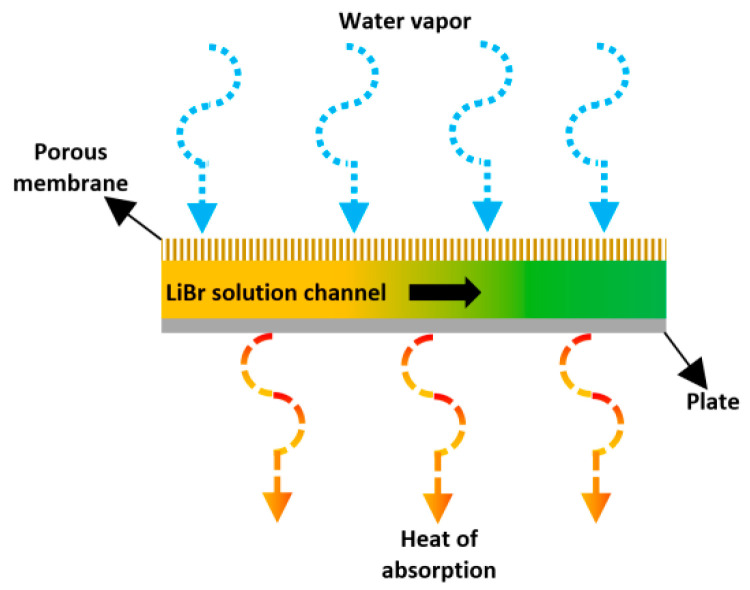
Schematic diagram of the absorber described by Isfahani et al. [52].

**Figure 10 membranes-10-00216-f010:**
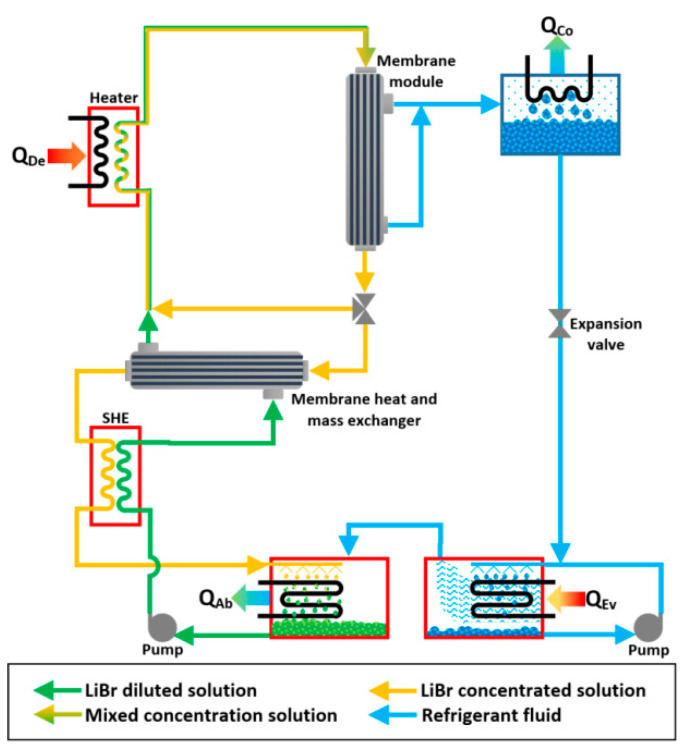
Membrane absorption system proposed by Hong et al. [131].

**Table 1 membranes-10-00216-t001:** Experimental work carried out on MD desorbers.

		Membrane Module Specifications					Solution Operation Parameters		
Authors	Working Mixture	Material	*A_m_* (cm^2^)	*δ_m_* (µm)	*d_m_* (µm)	*ϕ* (%)	ΔP (kPa)	T (°C)	X (% kg/kg)
RIffat et al. [48]	CHKO_2_-Water	Silicon	NA	NA	NA	NA	2.3–4.2	55 and 70	72 to 78
	CHKO_2_(55%) + CHCsO_2_ (25%)-Water							70	75
Thorud et al. [49]	LiBr-water	PTFE	16.8	142	0.45	NA	5.8 to 12.4	60	32 to 50
Wang et al. [51]	LiBr-water	PVDF	3000	150	0.16	85	NA	65 to 88	50
Isfahani et al. [52]	LiBr-water	PTFE	50.7	NA	0.45	85	13	70 to 120	50
Bigham et al. [53]	LiBr-water	NA	50.7	500	1	80	13 to 20	60 to 125	48 to 57
Isfahani et al. [54]	LiBr-water	PTFE	50.7	50	0.45	NA	3 to 20	50 to 125	48
Ibarra-Bahena et al. [61]	Carrol-water	PTFE	144	NA	0.22 and 0.45	NA	NA	66 to 84	60
Ibarra-Bahena et al. [62]	LiBr-water	PTFE	144	NA	0.45	NA	NA	75 to 95	45 to 60
Ibarra-Bahena et al. [64]	LiBr-water	PTFE	144	175	0.22	NA	NA	75 to 95	50
Venegas et al. [59]	LiBr-water	PTFE	87	NA	0.45	NA	NA	62 and 66	46
Ibarra-Bahena et al. [65]	LiBr-water	PTFE	144	NA	0.22	NA	NA	75 to 95	50

**Table 2 membranes-10-00216-t002:** Experimental studies carried out on membrane absorbers.

		Membrane Module Specifications					Solution Operation Parameters		
Authors	Working Mixture	Material	*A_m_* (cm^2^)	*δ_m_* (µm)	*d_m_* (µm)	*ϕ* (%)	ΔP (kPa)	T (°C)	X (% kg/kg)
Sudoh et al. [106]	LiBr-water	PTFE	23.8	80	0.2	75	4.6 to 6.4	30 to 73	35 to 55
Ali and Schwerdt [107]	LiBr-water	PTFE	46.6	60	0.22 to 1.0	60 to 85	0.9 to 2.3	24 to 29	50.8 to 54
Woods et al. [109]	LiCl-water	Oxyphan	6446.5	50	0.062	43	NA	16 to 37	30 to 39
	CaCl_2_-water							22 to 24	
Isfahani et al. [52]	LiBr-water	PTFE	77.1	NA	1	85	0.45 to 1.3	25 to 35	60
Isfahani and Moghaddam [110]	LiBr-water	NA	77.1	NA	1	80	0.4 to 1.3	25 to 35	60
Isfahani et al. [111]	LiBr-water	NA	74.1	NA	1	80	0.4 to 1.3	31 to 43	60
Huang [117]	LiCl-water	PVDF-Silica gel	2513	100	NA	NA	NA	35	55
Huang [118]	LiCl-water	PVDF/PVA	20944	150	0.15	15	NA	35	55
Berdasco et al. [119]	NH_3_-water	PTFE	138.7	60	0.05	70	NA	25	32
García-Hernando et al. [121]	LiBr-water	PTFE	109.2	175	1	85	1.5 to 3.0		58.6
				25–51	0.45	~90	1.5 to 2.0	26 to 28	60.0
				76–127	0.45	~90	~3.5		58.2
De Vega et al. [122]	LiBr-water	PTFE	109.2		0.45	~90	1.6 to 2.0	25.3	52.1

**Table 3 membranes-10-00216-t003:** Patents in membrane related absorption systems.

Inventor	Country	Year	Patent No.	Type of Patent	Cooling Type	Description
Victor A. Williamitis [134]	Canada	1972	US3645111	US	Expander	It was the combination of orifice and osmotic membranes. With the help of the heater, the pressure was raised in the condenser.
Raymond B. Trush [135]	USA	1991	US5024060	US	VCR	It was a modified compression refrigeration system using membrane and electrochemical compressor.
Johannes G. Wijmans, Richard W. Baker [136]	USA	1991	US5044166	US	VCR-VAR	The proposed design will be suitable for both the VCR and VAR refrigeration cycle with the help of the membrane unit.
John A. Broadbend [137]	USA	1994	US5329780	US	VCR	The flexible membrane was attached to the evaporator for producing the instant icing effect. The membrane was 0.003–0.005 inch made up of stainless steel.
Philip H. Coelho and Terry Wolf [138]	USA	1996	US5557943	US	Desiccant	Freezing at low cost was achieved by flowing the refrigerant via a thin film membrane. Even cryogenic temperature was possible. This was used for the storage of food and blood products.
Hans D. Linhardt and Joseph Rosener, Jr. [139]	USA	1999	US5873260	US	VAR	The membrane was used in the aqua-ammonia VAR system to improve the vapor quality of the refrigerant. Moreover, the dephlegmator/rectifier was not necessary, due to the presence of a membrane.
Anthony John Shacklock and Keith Brookes Spong [140]	New Zealand	1999	US5881566	US	VCR	The membrane tray was coupled with the heat rejecting device of compressor and condenser to improve the heat transfer rate.
Richard A. Callahan and Kishore V. Khandavalli [141]	USA	2000	US6128916	US	VCR	The non-condensable refrigerant from the condenser was removed and made to liquid with the help of a membrane. This will improve the quality of the refrigerant and avoids the vapor at the inlet of the capillary tube.
Wei Shyy, Marianne Monique Francois and Jacob Nan-Chu Chung [142]	USA	2003	US6598409B2	US	VCR	It was the combination of a VCR system and an ejector system. The flexible membrane also acted as a compressor and made the refrigerant vapor into drop by drop condensation process. It can also convert as a heat pump.
Hiromune Matsuoka, Kazuhide Mizutani, Nobuki Matsui And Manabu Yoshimi [143]	Japan	2007	US2007/0101759A1	US	VCR	The vacuum created by the non-condensable gas was removed by a membrane separation process. The membrane was placed next to the compressor, and the pure gas was condensed. The non-condensable gas was left to the atmosphere. Nonporous membranes like silicon rubber, polysulfone were used. It was best suitable for nitrogen and oxygen gas.
Manabu Yoshimi, Nobuki Matsui, Hiromune Matsuoka and Kazuhide Mizutani [144]	Japan	2007	US2007/0113581A1	US	VCR	Bridge circuit and a membrane separation were used to remove non-condensable gasses while installation, as well as during operations.
Manabu Yoshimi, Nobuki Matsui, Hiromune Matsuoka and Kazuhide Mizutani [145]	Japan	2008	US7357002B2	US	VCR	It was a combination of air-conditioning and VCR. The heat was used to separate the non-condensable gas on the refrigerant line via a separation membrane.
Matthias Seiler and Bernd Glockler [146]	USA	2010	US2010/0326126A1	US	VAR	The sorption process was integrated with the absorption process. The semipermeable membrane was used to separate the refrigerant from the liquid. It has a volatile refrigerant and non-volatile medium, which was having sorption pressure of 10^−6^ mbar at 20 °C. Best suitable for LiBr-Water mixture.
Saeed Moghaddam [147]	USA	2016	US9488392B2	US	VAR	The absorber and desorber were separated by a thin film layer, and the refrigerant was separated by the semipermeable membrane made up of nanofibers. In order to achieve the same efficiency in a conventional VAR system, a 2.7% high mass flow rate area was required. However, the advantage was less installation area and reduced components.
Wolfgang Heinzl [148]	Norway	2017	US9677791B2	US	VAR	The absorber and evaporator is a single component, and a generator and condenser as another component connected by a heat recovery unit. The semipermeable membrane was used to separate the generator and condenser, as well as the absorber and evaporator. Very low heat was sufficient to produce the refrigerant via the membrane.
Bamdad Bahar and Chunsheng Wang [149]	USA	2017	US2017/0362720 A1	US	VCR	The compressor in the VCR system was replaced by the electrochemical compressor. A mixture of ammonia and hydrogen was used as working fluid, and the remaining process was similar to the VCR system.
Rajiv Ranjan, Haralambos, Cordatos, Zissis A. Dardas, Georgios S. Zafiris, Yinshan Feng, Parmesh, Verma, Michael A. Stark [150]	USA	2020	US10584906B2	US	VCR	The permeable membrane was connected in the condenser of a VCR system to remove unwanted gas and contaminates. The vacuum pump was used to discharge or effective use of membrane separations.
Bo Jiang, Lei Wang, Haoquan Liu and Ruowu Xin [151]	China	2020	US2020/0064051A1	US	VCR and VAR	The oxygen inside the refrigerator was removed by an oxygen-enriched membrane. It keeps the food fresh for a long time.
Rajiv Ranjan, Yinshan Feng, Haralambos, Cordatos, Parmesh, Verma, Zissis A. Dardas [152]	US	2020	US2020/0149791A1	US	VCR	A low-pressure refrigerating system was designed with the help of double stage permeable membrane connected to the purge system.
Xiaobing Zhu, Bo Jiang, Lei Wang, Haoquan Liu and Ruowu Xin [153]	China	2020	US2020/0037640A1	US	VCR	The excess oxygen inside the refrigerator was removed by membrane assembly with the help of a pump. Hence, the maximum of air present with nitrogen was maintained inside the refrigerator. Hence, the wrinkles in the food were avoided, and the lifetime of the food stored in the refrigerator was increased.
Haoquan Liu, Bo Jiang, Lei Wang, Ruowu Xin [154]	China	2020	US2020/0064049A1	US	VCR	Up-gradation of patent no US2019/0301785A1. More space for storage of food with reduced oxygen.

USA, US: United States of America, VCR: Vapor Compression Refrigeration, VAR: Vapor absorption refrigeration.

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
