# Peer review of "Role of Membrane Technology in Absorption Heat Pumps: A Comprehensive Review"

_membranes, 2020, doi:10.3390/membranes10090216_

Round 1

Reviewer 1 Report

This study deals with a detailed review on the role of membrane technology in absorption heat pumps. The topic is interesting, and the manuscript is well-written. It can be acceptable for publication after following minor revisions:

  1. The outlook of the potential research directions should be carefully addressed in the study, which could offer a wider readership.
  2. Detailed comparisons of employing different membrane technologies in different components should be included, thus to address the key components of the system, where the performance could be more significantly improved via the membrane technologies.
  3. The copyright of the figures should be included.
  4. Instead of just listing the results of the previous literatures, some content should be written in a concise and sound way.
  5. Some recent studies on the MD process, such as “direct contact membrane distillation system for waste heat recovery: Modelling and multi-objective optimization” Analysis of temperature and concentration polarizations for performance improvement in direct contact membrane distillation” could be included to enrich the present study.

Author Response

Dear Reviewer 

Thank you very much for your comments and suggestions in order to improve de quality of the paper. 

Attached I am sending a document giving a response to your comments.

Yours sincerely.

Wilfrido Rivera 

On behalf of all the co-authors

Reviewer 2 Report

Authors give a comprehensive review on membrane separation in absorption heat pumps. It is an interesting and meaningful topic, and authors collected enough literatures for this review. Some comments for this review are listed:

  1. It is recomended to add the analyses or comparations on energy efficiency and economic efficiency in the paper;
  2. What is the main gauge for this different membrane systems, and which level that we have arrived now?
  3. It is better to add a paragraph to conclude the main limitations and what needs to be further developed in this paper.

Author Response

Dear Reviewer

Thank you very much for your comments and suggestions in order to improve the quality of the paper.

Attached I am sending you a document giving a response to your comments.

Yours sincerely,

Wilfrido Rivera 

On behalf of all the co-authors

Reviewer 3 Report

Introduction

The paper “Role of Membrane Technology in Absorption Heat Pumps: A Comprehensive Reviewis reviewed herein for its content, importance to the field, conciseness, style and clarity, and completeness.

Technical Content

The following technical questions/concerns should be resolved prior to publication:

  • Introduction, line 58: “…the absorption process of the ammonia vapor is the solution is exothermic…”. At this point, in the authors are introducing the absorption process with the refrigerant being described as the refrigerant. As the absorption process is exothermic by nature, why do the authors constrain their argument to ammonia? Should this not be left as the “refrigerant vapor”?
  • Section 2.1, line 170: “… and microchannels with 160 mm thick, 1 mm wide, and 38 mm wide…” The microchannels are per the reference, 160 μm thick, not 160 mm.
  • Section 3, line 482: The authors mention Oxyphan in the last sentence of this paragraph without any context, or previous mention. The referenced work mentions that using the membrane properties of Accural and an assumed of tortuosity 2 produces a membrane mass transfer coefficient 4.2 time greater than the membrane made from Oxyphan. The overall mass flux is only 2.1 time greater, due to the air flux. The author’s sentence does not reflect this and needs to be updated/corrected.
  • Section 4 Membrane-based heat Exchangers: The first few sentences of this section describe the need and use of solution heat exchangers in absorption cycles. These solution heat exchangers are almost always used to recover sensible heat within the cycle, and as such are not heat and mass exchangers (only heat exchangers). As such the balance of the section, which discusses membrane heat and mass exchangers is confusing. It’s not an issue about the information being wrong, but rather a question of whether or not it should be presented differently as the section does not seem cohesive.

Style and Clarity

The paper is laid out in a logical progression. While fairly well written there are several instances of spelling/grammatical mistakes, duplicate words, and poor/incomplete sentence structure that should be corrected. It is suggested that a comprehensive final read through by a technical editor be conducted.

Recommendation

The recommendation is that this paper should be published after revisions.

Author Response

(The authors gave the same response as above.)
